

# Generative artificial intelligence and machine learning methods to screen social media content

Kellen Sharp[1],*, Rachel R. Ouellette[2],*, Rujula Singh Rajendra Singh[3], Elise E. DeVito[2], Neil Kamdar[3], Amanda de la Noval[2], Dhiraj Murthy[4] and Grace Kong[2]

[1] Department of Radio-Television-Film, University of Texas at Austin, Austin, Texas, United States
[2] Department of Psychiatry, Yale School of Medicine, New Haven, Connecticut, United States
[3] Department of Computer Science, University of Texas at Austin, Austin, Texas, United States
[4] School of Journalism and Media, University of Texas at Austin, Austin, Texas, United States
* These authors contributed equally to this work.

Corresponding author
Dhiraj Murthy,
Dhiraj.Murthy@austin.utexas.edu

## ABSTRACT

**Background:** Social media research is confronted by the expansive and constantly evolving nature of social media data. Hashtags and keywords are frequently used to identify content related to a specific topic, but these search strategies often result in large numbers of irrelevant results. Therefore, methods are needed to quickly screen social media content based on a specific research question. The primary objective of this article is to present generative artificial intelligence (AI; *e.g.*, ChatGPT) and machine learning methods to screen content from social media platforms. As a proof of concept, we apply these methods to identify TikTok content related to e-cigarette use during pregnancy.

**Methods:** We searched TikTok for pregnancy and vaping content using 70 hashtag pairs related to "pregnancy" and "vaping" (*e.g.*, #pregnancytok and #ecigarette) to obtain 11,673 distinct posts. We extracted post videos, descriptions, and metadata using Zeeschuimer and PykTok library. To enhance textual analysis, we employed automatic speech recognition *via* the Whisper system to transcribe verbal content from each video. Next, we used the OpenCV library to extract frames from the videos, followed by object and text detection analysis using Oracle Cloud Vision. Finally, we merged all text data to create a consolidated dataset and entered this dataset into ChatGPT-4 to determine which posts are related to vaping and pregnancy. To refine the ChatGPT prompt used to screen for content, a human coder cross-checked ChatGPT-4's outputs for 10 out of every 100 metadata entries, with errors used to inform the final prompt. The final prompt was evaluated through human review, confirming for posts that contain "pregnancy" and "vape" content, comparing determinations to those made by ChatGPT.

**Results:** Our results indicated ChatGPT-4 classified 44.86% of the videos as exclusively related to pregnancy, 36.91% to vaping, and 8.91% as containing both topics. A human reviewer confirmed for vaping and pregnancy content in 45.38% of the TikTok posts identified by ChatGPT as containing relevant content. Human review of 10% of the posts screened out by ChatGPT identified a 99.06% agreement rate for excluded posts.

**Conclusions:** ChatGPT has mixed capacity to screen social media content that has been converted into text data using machine learning techniques such as object

detection. ChatGPT's sensitivity was found to be lower than a human coder in the current case example but has demonstrated power for screening out irrelevant content and can be used as an initial pass at screening content. Future studies should explore ways to enhance ChatGPT's sensitivity.

## INTRODUCTION

Social media is an evolving digital platform used widely. As of April 2024, 5.07 billion people use social media globally, which covers 93.2% of internet users (*Petrosyan, 2024*). Social media content can influence important health-related decisions by providing access to information and opinions from a variety of sources (*Iftikhar & Abaalkhail, 2017*). This is particularly concerning due to prevalent health-related misinformation identified on social media (*Wang et al., 2019*; *Suarez-Lledo & Alvarez-Galvez, 2021*; *Yeung et al., 2022*). However, social media can also be an effective pathway for increasing access to public health information (*Al-Dmour et al., 2020*; *Ghahramani, de Courten & Prokofieva, 2022*). To correct misinformation, it is first necessary to understand what content is being disseminated and shared on social media regarding a specific public health topic. The large scale and constantly evolving nature of social media requires the capacity to identify and characterize extensive amounts of content in a short period of time.

Our study aims to establish inroads between tobacco control and large language model (LLM) technologies, demonstrating a framework for collecting and analyzing large social media data corpora to generate insights for marginalized and/or targeted communities. Due to the rich and complex nature of social media content, qualitative analysis (*e.g.*, content analysis) is frequently used to characterize content in a specific topic area (*Ouellette, Selino & Kong, 2023*; *Kong et al., 2019*, *2021*; *Lee et al., 2023*). There has also been increased use of machine learning techniques to distill, describe, and examine large quantities of social media content (*Kong et al., 2023*). Machine learning methods are valuable for rapidly establishing a systematic and automated framework to analyze social media data (*Visweswaran et al., 2020*), including identifying the presence of specific objects (*e.g.*, *Murthy et al., 2024*; *Vassey et al., 2023*; *Wu et al., 2023*) and clustering data based on similar text and image content (*e.g.*, *Ketonen & Malik, 2020*; *Lee et al., 2024*). Advancements in artificial intelligence (AI), particularly generative AI, have empowered large language models to simulate human language processing capabilities (*Kalbit et al., 2024*), thereby expanding the scope and efficiency of large-scale social media research.

### Rationale for the case example topic: E-cigarette and pregnancy content on TikTok

In the current study, we examine the capacity of OpenAI's ChatGPT as a generative AI model to screen social media content from TikTok related to a salient public health topic:

use of e-cigarettes during pregnancy. We chose this topic due to limited research examining social media content in this topic area, despite its relevance to public health. This topic was chosen as an illustrative example, but our methods can be applied across a broad range of other topics on social media.

Electronic cigarettes (*i.e.*, e-cigarettes) have been the most frequently used nicotine/tobacco product among youth and young adults for the past decade (*National Center for Chronic Disease Prevention and Health Promotion (US) Office on Smoking and Health, 2016*; *Cullen et al., 2018*; *Cornelius et al., 2020*), many of whom are now reaching child-rearing age. E-cigarette use among pregnant people was selected as a case example because it is a timely topic with clinical and public health relevance. Combustible cigarette smoking during pregnancy is linked with clear and well-substantiated risks for adverse health outcomes for the pregnant person and baby, as communicated to the public through effective public health messaging and advice from healthcare providers (*Office of the Surgeon General (US) & Office on Smoking and Health (US), 2004*; *Wickstrom, 2007*; *Allen, Prince & Dietz, 2009*; *Einarson & Riordan, 2009*; *Himes et al., 2013*; *Allen, Oncken & Hatsukami, 2014*; *National Center for Chronic Disease Prevention and Health Promotion (US) Office on Smoking and Health, 2014*; *Allen et al., 2018*; *Gould et al., 2020*; *Morales-Prieto et al., 2021*).

In contrast, the potential impacts of e-cigarette use during pregnancy are less thoroughly understood, with mixed findings from human studies as to the risks of e-cigarette use during pregnancy, including from clinical trials, pregnancy cohort studies, and population survey data (*DeVito et al., 2021*). Although the data is still incomplete, the current available evidence (clinical, epidemiological data, preclinical, and *in vitro* data) taken together indicate that e-cigarette use (exclusive e-cigarette use or dual use of e-cigarette and combustible cigarettes) during pregnancy most likely poses greater risk than abstaining from all tobacco products, but exclusive e-cigarette use may be less harmful than smoking combustible tobacco. These mixed findings can complicate the development and dissemination of cohesive messaging on platforms such as social media. Analysis of social media content created by the general public can also provide a snapshot into the public's current understanding and opinions about the use of e-cigarettes during pregnancy.

### E-cigarette content on TikTok

Recently, studies have focused on TikTok as a social media platform which is growing in popularity, serving as the second most popular social media platform among young adults ages 18–29 in 2023 (*Dixon, 2024*; *Pew Research Center, 2023*). Portrayal of e-cigarettes is prevalent on TikTok, with more than half of all e-cigarette content identified as "promoting vaping" (*Basch et al., 2021*). Pro-e-cigarette content previously identified on TikTok includes videos featuring vaping tricks, customization, advertisement, and trendiness (*Xie et al., 2023*). TikTok has been examined as a platform to communicate and surveille the adverse effects of nicotine on people's health (*Jancey et al., 2023*), informing and supporting counter-messaging efforts. TikTok has therefore been found to contain a mixture of both pro- and anti-e-cigarette content. Other studies have focused on the need

for greater surveillance and regulation within TikTok, pointing to the platform's lack of e-cigarette content moderation, resulting in trending e-cigarette-related hashtags for vape products eliciting millions of views (*Tan & Weinreich, 2021*).

As it relates to e-cigarette use during pregnancy, one study assessed online forums for pro- and anti-vaping content, including misinformation, related to pregnancy (*Wigginton, Morphett & Gartner, 2016*). However, no studies have assessed content related to e-cigarette use during pregnancy on TikTok. To understand what content and information about e-cigarette use during pregnancy is being shared on platforms such as TikTok, we first need to identify relevant content.

## Power of combining machine learning with generative AI to screen social media content

Machine learning methods are valuable for establishing a systematic and automated framework to analyze data from social media platforms (*Visweswaran et al., 2020*). While machine learning can examine large amounts of data quickly, datasets typically still require human review. For example, unsupervised models such as clustering techniques can distill large amounts of data into fewer parts (*i.e.*, clusters) for further analysis. However, clusters become uninterpretable if there is too much noise resulting from irrelevant data. Therefore, human review is typically needed to ensure that posts include content related to the primary topic(s) of interest (*e.g.*, e-cigarettes and pregnancy).

Additionally, traditional machine learning classifiers can be stringent and rely on a finite dictionary list of words, allowing limited flexibility in text data classification through processes like stemming and lemmatization. Similar limitations exist for screening methods such as keyword filtering, which requires pre-identification of a comprehensive set of keywords to identify relevant content. Due to the rapidly evolving nature of social media, the terms used to describe topics such as vaping and pregnancy are constantly changing, calling for more flexible approaches that look for patterns in text rather than discrete terms. Human review of content is also frequently used to screen for relevant content, particularly for complex topics where computational methods such as keyword filtering may miss content. However, human review can be a time consuming and cumbersome process for large-scale social media datasets.

Advancements in artificial intelligence, as seen in generative AI, present exciting new opportunities to apply data science techniques to both identify and characterize social media content. Recent developments in AI technology have broadened the scope and influence of large-scale social media research. For example, ChatGPT (*OpenAI, 2023*) is an LLM trained on a vast web *corpus* that has demonstrated state-of-the-art performance on a range of natural language tasks (*Kasneci et al., 2023*). ChatGPT's models, including GPT-4, are trained on a diverse array of sources including books and websites. Its transformer architecture and advanced natural language processing capabilities can be leveraged to efficiently analyze patterns across data and make relevance assessments (*OpenAI, 2023*). ChatGPT's natural language capacity can eliminate the need for manual screening and pre-training for concepts like e-cigarettes and pregnancy, instead using existing definitions informed by ChatGPT's vast *corpus* of data, elicited using specific prompts.

Prompt engineering is a technique for developing and optimizing requests to elicit a desired output from LLMs such as ChatGPT (*Brown et al., 2020*). In an application of ChatGPT's capabilities, *Li et al. (2024)* utilized the LLM through prompt engineering on a dataset consisting of 3,481 hateful, offensive, and toxic (HOT) comments posted on Reddit, X (formerly Twitter), and YouTube. Their study aimed to compare ChatGPT outputs to a previous coding analysis conducted by human coders with the same dataset. They observed that ChatGPT was generally reliable and consistent, achieving an 80% accuracy rate in comparison to human coders. ChatGPT therefore presents unique potential to streamline social media research by facilitating quicker screening of content as relevant or irrelevant to a specific topic. In our current study, we examine ChatGPT's capacity to screen TikTok posts for e-cigarette and pregnancy-related content.

However, much is still unknown about ChatGPT and its potential limitations when screening social media content. LLMs like ChatGPT can experience a loss of dialogue or "contextual forgetting" in conversational threads due to limitations in memory processing and retaining extensive context in conversations (*Liang et al., 2023*). Furthermore, previous work has found that ChatGPT can produce erroneous outputs or "hallucinations" (*Alkaissi & McFarlane, 2023*). Therefore, further comparison between ChatGPT and other commonly used methods, such as screening by human coders, is needed to evaluate ChatGPT's capacity to identify specific types of content and topics.

## Machine learning techniques to prepare social media data for ChatGPT

Another important limitation of ChatGPT in its current form is its limited capacity to examine videos, a format that is central to popular platforms such as TikTok. ChatGPT has begun to incorporate more image and video analysis features. However, ChatGPT's capacity to examine text still far exceeds its capacity to examine image-based content. To overcome this obstacle, machine learning techniques such as computer vision and natural language processing can facilitate the conversion of video-based content into written transcripts, facilitating text-based screening of social media content by ChatGPT.

Previous work (*Pinto et al., 2024*) on social media has taken multimodal approaches to analyze online communities, including using data scraping tools like Zeeschuimer and PykTok to extract community metadata from TikTok (*Zehrung & Chen, 2024*; *Van Berge, 2023*). The multimodal approaches used in this study extend these methods by including the acquisition of audio, image, and text data from TikTok posts, converting the audio and image data into text, and using ChatGPT to identify content related to vaping and pregnancy. To convert video-based social media content into text-based data, we developed a multimodal system that automates the extraction and integration of image and audio content into text using computer vision techniques (*i.e.*, object detection and optical character recognition) and natural language processing (*i.e.*, automatic speech recognition).

One computer vision technique that is ideal for converting image data into text is object detection. Object detection is a computer vision technique that identifies and locates objects of interest within images or videos by drawing bounding boxes around the detected

objects and classifying them into predefined categories (*Mack et al., 2008*; *Xiao et al., 2020*). State-of-the-art object detection algorithms leverage deep learning, feature extraction and integration, and machine learning techniques to accurately recognize and locate objects with minimal training (*Xiao et al., 2020*; *Wang, Bochkovskiy & Liao, 2023*). Recent studies have employed object detection technology to accurately identify e-cigarette content on TikTok, detecting the presence of objects such as vape devices and vapor clouds (*Murthy et al., 2024*), as well as e-cigarette containers, brand names, and warning labels (*Vassey et al., 2023*). Additionally, previous research has leveraged deep learning technology to identify the presence of e-cigarette devices (*Vassey et al., 2020*) and FDA warning labels (*Kennedy et al., 2021*) in Instagram posts. By applying object detection techniques, we can therefore identify posts that are likely to include e-cigarette content by detecting specific objects (*e.g.*, e-cigarettes) associated with specific topics.

In addition to objects, TikTok posts frequently include text incorporated within video content. Optical character recognition (OCR) is a type of computer vision technique that converts images of text into machine-readable and editable text data (*Hamad & Kaya, 2016*). The process of OCR involves detecting, recognizing, and extracting characters, words, and lines of text from images or scanned documents. OCR algorithms may employ image preprocessing, feature extraction, and machine learning techniques to accurately recognize and extract textual information (*Hamad & Kaya, 2016*; *Memon et al., 2020*). OCR may be used in social media research to enable the extraction of textual data from images. Previous work (*Salma et al., 2021*) has used both object detection and OCR computer vision techniques to detect and extract relevant information from visual data, such as license plates. When applied to social media, the employment of both object detection and OCR techniques can facilitate the conversion of image-based information into text-based data. This text-based data can then be entered into ChatGPT to screen for specific topics such as vaping and pregnancy.

Finally, TikTok posts often include an individual or individuals talking to the camera, resulting in large amounts of relevant information being communicated through audio. Automatic speech recognition (ASR) is a natural language processing technology that enables computers to process and convert human speech into text. Modern ASRs, like OpenAI's Whisper (*OpenAI, 2022*), use end-to-end neural network models that can directly map audio waveforms to text, bypassing the intermediate phonetic elements used by traditional ASR systems (*Ghai & Singh, 2012*). Previous research has showcased Whisper's potential in health communication by providing a tutorial for the ethical implementation of the software in an audio transcription pipeline for psychiatry, psychology, and neuroscience research (*Spiller et al., 2023*). Moreover, advancements in deep learning technology have greatly refined ASR accuracy as evidenced by studies examining its improving reliability to capture different languages (*Mukhamadiyev et al., 2022*) and accents (*Najafian & Russell, 2020*). ASRs such as OpenAI's Whisper can therefore facilitate the conversion of speech to written text for further analysis by ChatGPT.

## Current study

The primary objective of this study is to develop a multimodal system for converting video-based content into text data, with the resulting text data entered into ChatGPT to examine its capacity to screen TikTok content for specific topics (*i.e.*, e-cigarettes and pregnancy). Our multimodal system automates the extraction and integration of comprehensive information from videos into a large, text-based dataset using machine learning techniques including object detection, optical character recognition, and automatic speech recognition. We then developed a prompt procedure for identifying TikTok posts containing e-cigarette and pregnancy content, comparing ChatGPT's determinations to a human coder to better understand its strengths and limitations when screening content. Our research questions include:

*Research Question 1: How can machine learning be used to develop a multimodal system that automates the extraction and integration of comprehensive text-based data from videos in a large dataset of TikTok posts?*

*Research Question 2: What is ChatGPT's capacity and accuracy, as compared to human coding, when screening large social media datasets for two content areas simultaneously (i.e., e-cigarettes and pregnancy)?*

# MATERIALS AND METHODS

## Overview of data collection, aggregation, and analytic methods

We obtained TikTok metadata from $N = 23,888$ videos using Zeeschuimer (*Digital Methods Initiative, 2023*), a scraping tool. After a deduplication process, we identified $N = 11,673$ unique videos. We employed the PykTok library (*Freelon, 2022*) to download video content for these 11,673 posts using their video IDs. We then used Whisper (*OpenAI, 2022*) to transcribe audio content from each video. Our multimodal approach then integrated computer vision techniques to create a comprehensive text-based dataset. Specifically, we used the OpenCV library (*Bradski, 2000*) to extract still frames of each video. After extracting still frames, we used the AI service Oracle Cloud Infrastructure (OCI) for image processing (*Oracle Corporation, 2025*). OCI's module, Oracle Cloud Vision, is an AI service used to conduct extensive deep-learning-based image analysis. This module leverages OCR to extract and analyze textual information. These methods are summarized visually in Figs. 1A–1G, detailing the sequential steps involved in our data collection and data compilation process. Finally, prompts were generated through an iterative process in ChatGPT-4 to screen the resulting text-based datasets for content discussing e-cigarettes and pregnancy. Additionally, we conducted frequency analyses of the objects and texts detected within posts, enriching our understanding of the dataset. These analytical methods are summarized visually in Figs. 1G–1L.

Data processing steps are illustrated in panels a–f, resulting in a processed dataset (j) comprised of three key components (g–i), which were fed to ChatGPT for classification into vaping, pregnancy, or both vaping and pregnancy categories, represented by (k). Additionally, the objects and text detected in posts (i) underwent frequency analysis (l) through Oracle Cloud Vision.

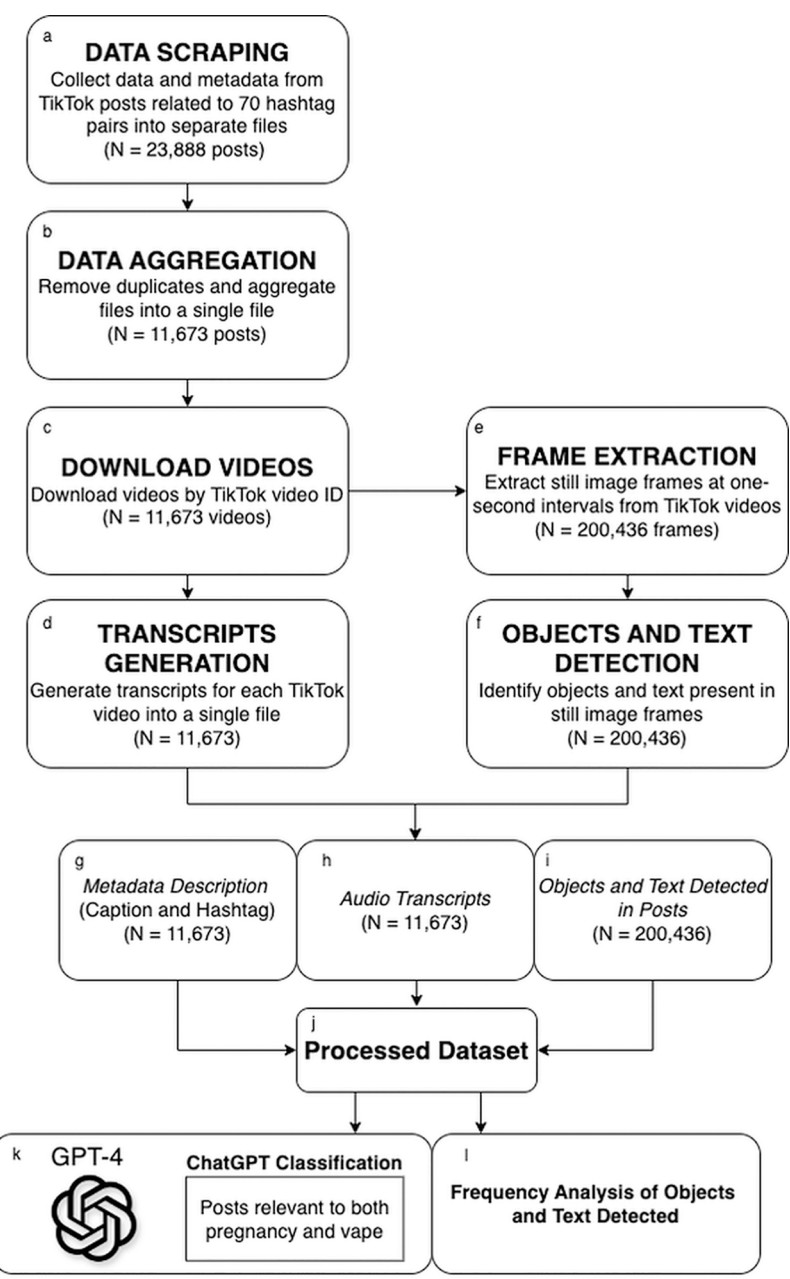

**Figure 1 Multimodal data collection and dataset compilation workflow for TikTok content.** Data processing steps are illustrated in panels (A–F), resulting in a processed dataset (J) comprised of three key components (G–I), which were fed to ChatGPT for classification into vaping, pregnancy, or both vaping and pregnancy categories, represented by (K). Additionally, the objects and text detected in posts (I) underwent frequency analysis (L) through Oracle Cloud Vision.

Our research methods are informed by the Association of Internet Researchers (AoIR) research integrity guidelines including providing a detailed methodology and transparent reporting of results, pursuing review by an ethics committee or institutional review board

(IRB), and minimizing use and sharing of identifiable information (*Franzke et al., 2020*). Specifically, the Yale Institutional Review Board determined the work to be non-human research and therefore exempt from further IRB review. No images were input into ChatGPT to minimize use of identifiable information, and only publicly available content was used for analysis.

## Data collection

We searched for e-cigarette and pregnancy-related content on TikTok using 70 pairs of thematic hashtags related to "pregnancy" (*e.g.*, #pregnancytok, #momtok, #pregnancy) and "vape" (*e.g.*, #vape, #smoke, #juul; see Appendix A, Table S1). These hashtags were identified by starting with six broad keywords related to both pregnancy and vaping ("e-cigarette pregnant", "e-cigarette pregnancy", "vaping pregnant", "vaping pregnancy", "vape pregnant", "vape pregnancy"). We then examined the hashtags used in posts containing both vaping and pregnancy content to identify the hashtags and hashtag pairs used most frequently across relevant posts. A total of 70 pairs of hashtags were identified as being related to vaping and pregnancy, reflected in Appendix A, Table S1.

We commenced data collection in February 2024 using Zeeschuimer (*Digital Methods Initiative, 2023*), a browser extension optimized for scraping data from social media platforms, including TikTok. Following methods employed by previous researchers (*Van Berge, 2023*), we used Zeeschuimer to scrape TikTok data by manually scrolling through the site interface. The Zeeschuimer extension exports scraped metadata in Newline Delimited JSON (NDJSON) format, which is a structured file format commonly used for storing extracted data.

This dataset ($N$ = 23,888) contained unique identifiers (*i.e.*, IDs), titles for each video, video descriptions provided by the content creator, and engagement metrics (*e.g.*, the number of likes, shares, comments, and overall view counts) (see Fig. 1A). To ensure data uniqueness in our extensive dataset, we conducted a deduplication process, by identifying and merging duplicate entries based on the unique video IDs, successfully reducing our dataset to 11,673 unique, unprocessed video metadata entries (see Fig. 1B).

We used PykTok library (*Freelon, 2022*) to download the identified TikTok videos (see Fig. 1C). The PykTok module retrieves data directly from TikTok web pages and undocumented application program interfaces (APIs), enabling the download of videos and metadata for local storage. PykTok streamlined the download process by generating direct URLs for each identified video, which we used to acquire video content corresponding to the cleaned dataset of video IDs.

## Dataset preparation using machine learning

### Automatic speech recognition

For each of the 11,673 unique TikTok videos, we first generated transcripts of the audio using the ASR system, Whisper (see Fig. 1D). Whisper is an advanced automatic speech recognition tool that can efficiently transcribe verbal content embedded within videos

(*OpenAI, 2022*). Previous studies have used Whisper in conjunction with PykTok to enhance TikTok metadata collection (*Pinto et al., 2024*).

### Computer vision

To identify and extract image content, including objects and text, from TikTok videos, we used a Python script with the OpenCV library (see Fig. 1E) to extract frames from the unprocessed 11,673 downloaded videos. The script was programmed to capture one frame per second from each video, maximizing the amount of information captured. This resulted in a total of 200,436 unique frames (an average of 17 frames per video) that were used for the detection of objects and text (see Fig. 1F).

We examined the 200,436 frames using OCI Vision, an AI service for image-based deep-learning image analysis (*Oracle Corporation, 2025*). OCI Vision has prebuilt models that uses OCR to detect objects and classify images, providing confidence levels for detected elements. OCI Vision recorded attributes such as detected faces and details about the AI model version used in the analysis, including the version number and updates. We focused on the most frequently detected objects and words across different video categories, specifically targeting the top 20 in each category. This allowed us to examine the most common objects (*e.g.*, "person," "human face," and "clothing,") and words presented in text overlays (*e.g.*, "smoke," "baby," and "health") present across frames and videos. The identified text and objects were combined across image frames for each video, resulting in one list of objects and text per post.

### Dataset compilation

After completing the ASR and object detection analyses, the resulting data (*i.e.*, transcripts and list of detected objects and text) were combined with the post description and hashtags into one dataset (Figs. 1G–1I). The dataset was formatted as a CSV file, with each row representing one video and columns to reflect each category of data, including the video ID, post metadata description and hashtags; (Fig. 1G), ASR-generated transcript (musical and spoken content in any language; Fig. 1H), and list of computer-vision detected text and objects (Fig. 1I). This comprehensive approach allows ChatGPT-4 to acquire deeper insights into the video content, enhancing its ability to accurately determine relevance to topics pertaining to both pregnancy and e-cigarette use.

## Data screening using ChatGPT

### ChatGPT prompt generation

Prompt engineering is becoming an established methodological tool that, in recent years, has been used to advance healthcare research by improving question-answering systems, text summarization, and machine translation (*Wang et al., 2024*). We continued this trend of health-related prompt engineering by applying it to ChatGPT-4 with our consolidated dataset. As we developed our study's methodology, our instructional prompt for ChatGPT underwent multiple iterations similar to the approach used by *Li et al. (2024)*. This iterative prompt engineering process enabled us to experiment and refine until we reached a point where ChatGPT understood our desired output. In total, we conducted four iterations. For

each iteration, a study team member reviewed ChatGPT's output for ten out of every 100 posts. The prompt was then further refined to help increase the accuracy, interpretability, and reliability of ChatGPT's output. Below we detail our process for refining our ChatGPT prompt across the four iterations until we identified the final prompt.

Our initial prompt was, "*List whether each video talks about pregnancy or vaping.*" This yielded excessive rationale for its interpretation of classifications, making large scale analysis difficult. To address this challenge, we revised the prompt to a simpler format, "*For each video, indicate 'true' or 'false' for the presence of content related to pregnancy and vaping.*" While streamlining outputs, it resulted in unreliable explanations, hovering around potential true and false categorizations for all content. We discarded "true and false" metrics and refined our prompt to "*For each video, provide a structured entry: video_id, relevant to pregnancy (yes/no), relevant to vaping (yes/no).*" Although this prompt was effective for ensuring data entry uniformity, it lacked structural output clarity. Finally, we optimized the prompt to address issues of unclear and non-concise output and provide a navigable structure. "*For each video provided, classify and record in a table with the following columns: video_id, relevance to pregnancy (yes/no), relevance to vaping (yes/no).*" Vaping and pregnancy content was therefore searched for separately by ChatGPT. Videos indicated as having both vaping and pregnancy related content were identified using an "AND" operation in the analytic process after ChatGPT screened for relevant content.

### ChatGPT screening process

We used the GPT-4 model to screen the full dataset of $N = 11,673$ videos by attaching the csv file with the consolidated dataset described in the **Data Compilation** section above and entering the final prompt. ChatGPT then generated a new csv file formatted as indicated in the prompt *i.e.*, with columns for the video_id, relevance to pregnancy (yes/no), and relevance to vaping (yes/no).

### Human verification

To examine the accuracy of ChatGPT's classifications, we implemented a manual human coding process. A human coder viewed all TikTok posts marked by ChatGPT as relevant to both pregnancy and vaping by watching the video and reading the post description. A human coder also reviewed 10% of the posts identified by ChatGPT as not relevant to pregnancy and vaping, as done frequently in studies with multiple qualitative coders to examine media content (*Buckton et al., 2018*; *Lee, Murthy & Kong, 2023*; *Manganello, Franzini & Jordan, 2008*). Posts were not indicated as including both pregnancy and vaping content if this content was present in the hashtags only (*e.g.*, includes #vape in the list of hashtags but does not discuss e-cigarettes further in the video or post description). Posts were also excluded if they discussed use of nicotine or tobacco (*e.g.*, smoking) more broadly without explicit discussions of vaping and/or e-cigarettes. This human verification process was to evaluate the accuracy of ChatGPT's determinations and our use of prompts in identifying content relevant to e-cigarettes and pregnancy.

# RESULTS

## ChatGPT relevance classifier results

The ChatGPT Relevance Classifier analysis identified TikTok posts related to both vaping and pregnancy, using the capabilities of ChatGPT-4 through prompt engineering (see Fig. 1K). ChatGPT determined that 44.86% ($N$ = 5,237) of the screened videos were relevant to pregnancy only, while 36.91% ($N$ = 4,311) of videos pertained to vaping only. Only 8.91% of the dataset ($N$ = 1,040) contained videos that ChatGPT determined were relevant to both pregnancy and vaping.

Figure 2 features a still where both "pregnant" and "vape" appear in the textual content. The transcript and captions explicitly mention pregnancy and vaping, highlighting the classifier's ability to accurately identify and categorize posts that intersect these two topics. In cases where the video only mentioned topics related to "pregnancy" and "vape," ChatGPT would generate a "no" response.

## Object detection results

To further illustrate how the objects and text were detected, we examined the frequency of objects detected across the 1,040 posts identified by ChatGPT as relevant. OCI Vision identified over 75% of the videos as featuring people (86.54%), human faces (38.46%), and clothing (36.06%), with the potential for overlapping categories. Moreover, women (31.25%) were detected in over three times as many videos as men (9.62%). Other less frequently detected objects included accessories (17.31%), furniture (8.65%), and vehicles (5.77%) (see Fig. S1). The text detected in videos covered topics across vaping and pregnancy (see Fig. S2). The most prevalent term, "pregnant," appears in approximately 26.44% of the videos. Terms directly related to smoking, such as "smoking" (19.23%), "smoke" (11.54%), "vaping" (14.42%), and "vape" (13.94%) were featured frequently. "Pregnancy" is mentioned in 16.35% of the videos, "baby" is present in 13.94%, and "mom" is found in 7.21% of the videos. Action-oriented terms such as "quit" (11.54%), "like" (11.35%), "get" (8.65%), and "stop" (7.21%) were also present.

## Human verification results

To further validate the accuracy of ChatGPT's classifications, a human coder cross-checked the 1,040 posts that ChatGPT marked as relevant to both pregnancy and vaping. Upon evaluation, the coder verified $n$ = 861 posts (82.79% of the 1,040 posts identified by ChatGPT) as containing pregnancy content, $n$ = 555 posts (53.37%) as containing e-cigarette content, and $n$ = 472 posts (45.38%) as discussing e-cigarette use during pregnancy.

A human coder also reviewed 10% of the 10,633 posts that ChatGPT determined to not be relevant to both pregnancy and vaping. Out of the 1,063 human-reviewed posts, $n$ = 10 (0.94%) were determined to actually include pregnancy and vaping content (*i.e.*, were misclassified by ChatGPT as irrelevant), resulting in 99% agreement between ChatGPT and the human coder for screened-out posts.

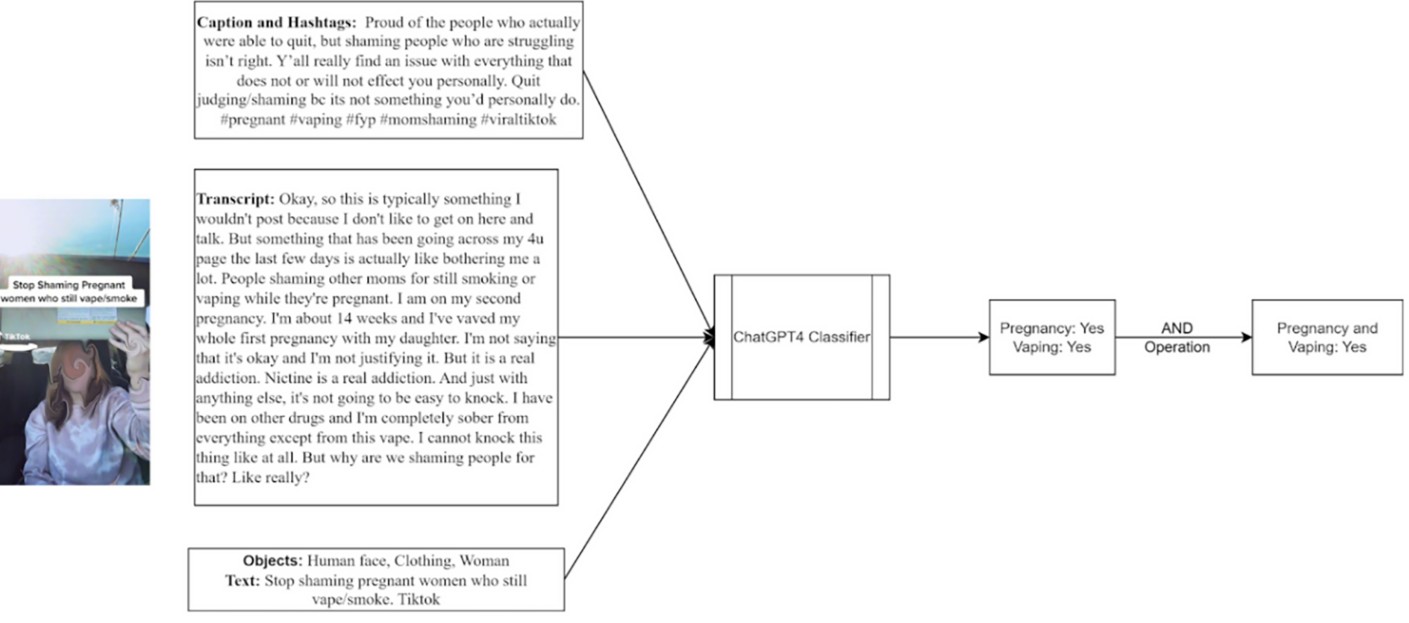

**Figure 2** **Illustration of ChatGPT screening process examining for pregnancy and vaping content.** Figure 2 features a still where both "pregnant" and "vape" appear in the textual content. The transcript and captions explicitly mention pregnancy and vaping, highlighting the classifier's ability to accurately identify and categorize posts that intersect these two topics. In cases where the video only mentioned topics related to "pregnancy" and "vape," ChatGPT would generate a "no" response.

# DISCUSSION

Frequently used methods for examining social media content, including machine learning methods such as object detection and image or text-based clustering, typically require human verification of relevant content prior to analysis. This can be a cumbersome and time intensive process due to the expansive and constantly evolving nature of social media content. Generative AI tools such as ChatGPT can facilitate labor-intensive processes to enhance computational abilities. However, these capabilities are still at a nascent stage. Our methods align with methods used by *Trajanov et al. (2023)* and *Li et al. (2024)*, who combined the power of LLMs like ChatGPT with human oversight to examine the potential limits of ChatGPT in identifying and categorizing social media content.

Prior use of generative AI to screen and code social media content has focused on text-based media sources (*Li et al., 2024*). We expanded these methods by developing a multimodal system using machine learning methods (*i.e.*, object detection, optical character recognition, and automatic speech recognition) for converting video-based content into text-based data to be input into ChatGPT. We then examined ChatGPT's capacity to screen a large, text-based dataset of TikTok posts for e-cigarette and pregnancy content. We leveraged ChatGPT, using its Generative Pre-trained Transformer 4 (GPT-4) model, to examine 11,673 distinct data entries, including one data entry per social media post containing the TikTok post description, transcribed audio, and identified objects and text. Each data entry was screened by ChatGPT for content relevant to either pregnancy,

vaping, or both, resulting in 1,040 out of 11,673 posts identified by ChatGPT as containing both pregnancy and vaping content.

A human coder then reviewed all 1,040 posts for relevant content, identifying 472 posts discussing both e-cigarettes and pregnancy, for a 45.38% concordance rate. These discrepancies may be due to multiple reasons. When generating prompts in ChatGPT, it was not able to accurately interpret prompts that searched for the intersection of two topic areas (*e.g.*, "*For each video, indicate 'true' or 'false' for the presence of content related to pregnancy and vaping.*"). To account for this limitation, we had to search for vaping and pregnancy content separately (*i.e.*, "*For each video provided, classify and record in a table with the following columns: video_id, relevance to pregnancy (yes/no), relevance to vaping (yes/no).*"), and then detected for intersections in content (*e.g.*, discussions about e-cigarette use during pregnancy) by identifying posts that discussed both vaping and pregnancy. The human verification process also included a few unique elements that may have resulted in higher sensitivity including: 1) review of the actual TikTok video which may have included additional information that could not be converted into text using our machine learning multimodal process; 2) more strict definitions of relevant content including exclusion of content that only mentioned one of the topics in the hashtags without inclusion of relevant content in the post video or description; 3) greater ability to decipher between discussions of nicotine vaping (*i.e.*, e-cigarette use) *vs.* cannabis vaping.

E-cigarettes are rarely called e-cigarettes on social media, with colloquial terms such as "vaping" being more prevalent. This informed our search strategy and decision to use the word "vaping" in the ChatGPT prompt instead of "e-cigarettes". However, increased popularity of cannabis vaping has resulted in similar hashtags (*e.g.*, #vape) and words (*e.g.*, "vaping") being used in both e-cigarette and cannabis vaping social media content. Human verification may be better equipped to identify and decipher these nuanced details in content discussing complex social constructs such as vaping on social media. ChatGPT was more accurate and effective at screening out irrelevant content. After human review of 10% of the posts deemed by ChatGPT to not include pregnancy and vaping content, only 0.94% of those screened-out posts were determined to be misclassified as irrelevant. Therefore, our findings indicate that ChatGPT's determinations may be higher in specificity (*i.e.*, ability to screen out irrelevant posts) than sensitivity. Meanwhile, determinations made by human coders may be higher in sensitivity (*i.e.*, ability to identify the presence of more nuanced content) but are limited in their capacity to screen large numbers of social media posts quickly. The capacity of ChatGPT to narrow 11,673 posts down to 1,040 results for human review still supports the potential use of generative AI platforms, including LLMs such as ChatGPT, to streamline the identification of relevant social media content. However, our findings also support the continued need for human oversight, particularly when examining for the intersection between two complex topics such as pregnancy and e-cigarettes.

At scale, ChatGPT's strength lies in its ability to quickly parse through vast amounts of data. Scholars (*Zhang et al., 2023*) describe the power of LLMs like ChatGPT for extracting key insights from vast amounts of text, akin to "finding a needle in a haystack." Our results

suggest that ChatGPT may not yet be as advanced at identifying the "needle in the haystack," but does exhibit proficiency in identifying the *correct pile of hay*. The accuracy of ChatGPT's classifications highlights its broader categorization approach, indicative of differences in specificity rather than "hallucination." Therefore, ChatGPT's determinations may be particularly helpful for weeding out posts that are highly irrelevant, which can provide human verifiers with more streamlined datasets for more targeted review and coding. Through prompt engineering, researchers can also iteratively customize ChatGPT to output more accurate and specific answers based on highly tailored inputs (*Meskó, 2023*). To improve concordance between ChatGPT and human verification in future studies, additional definitions, examples, and key terms (*e.g.*, differentiating cannabis and nicotine vaping content) can be included in the ChatGPT prompt using a few-shot learning approach (*Brown et al., 2020*).

Our two-step approach—first using an LLM and then cross-checking its outputs with human annotations—echoes sentiments by *Reiss (2023)*, who describes ChatGPT as a "nondeterministic" tool that requires human oversight. This model ensures the dataset remains robust and provides a foundation for in-depth analysis, while human oversight refines and verifies the dataset. Moreover, the integration of human oversight ensures that the classifications are not only consistent with computational analysis but also resonate with human judgment and interpretation of social constructs and behaviors such as vaping, providing a layered approach to validation. This multimodal and mixed-methods approach validates the strength and reliability of the findings, making a significant contribution to new research methodologies in the analysis of social media content.

A novel methodological aspect of our study is the enrichment of a social media dataset with image classification labels, detected text, audio transcriptions, and ontology classes which enhanced semantic categorization and analysis. By incorporating details about object and text detection model versions, our model ensures transparency and reproducibility in the analysis process. Our study also used computer vision techniques to detect and classify objects and text within our metadata entries. The prevalence of human elements in the object detection results suggests an attempt to engage viewers on an emotional and personal level when discussing the impacts of vaping during pregnancy. Terms directly related to the topic of smoking, such as "smoking" (19.23%) and "smoke" (11.54%), are also notably frequent, underscoring the emphasis on smoking behaviors, their impacts, and the potential effects of switching from smoking to vaping during pregnancy. This is consistent with data showing that many people who use e-cigarettes during pregnancy either use e-cigarettes alongside cigarettes or use them in place of cigarettes, and report using e-cigarettes in an attempt to either cut down on their cigarette smoking or as a smoking cessation aid. However, future studies are needed to analyze the content of these videos.

## Limitations and future directions

Our work has several limitations. We converted all image and audio data into text using machine learning techniques because, at the time, ChatGPT had limited capacity in image

processing. As LLMs are in their nascent stage of image processing, future studies may not need to preprocess images and videos using computer vision models. Nonetheless, sensitivity analyses are still needed comparing the accuracy of ChatGPT's determinations based on visual (*i.e.*, videos and images) *vs.* text-based (*i.e.*, transcripts) inputs.

Additionally, even if ChatGPT and other generative AI tools improve their abilities to assess and analyze images and videos, there could be issues of privacy in inputting sensitive information such as faces. Therefore, computer vision techniques that can convert image and video-based data into text may still have utility. Our team did not enter image data into ChatGPT and only used text from publicly available TikTok posts. However, all use of generative AI models and platforms such as ChatGPT for research purposes should be done carefully, with effort to minimize the use of private and identifiable information, as data inputted into ChatGPT can be used to train its model. For future works, analyzing the objects and text commonly found in relevant posts, contrasted with those in irrelevant posts, could help discover patterns or features that distinguish meaningful content from noise. Comparisons in objects between relevant and irrelevant content can also identify common and unique objects present in posts discussing specific topics.

Other LLMs (*e.g.*, Microsoft's large language model—Copilot) could provide additional insights into how pregnancy and vaping content can be classified. Sensitivity analyses should assess which tools provide the most accurate determinations, given that all LLM service algorithms are proprietary. Other models can be used for cross-validation in future work (*e.g.*, Claude, Gemini, LLaMA), as well as different versions of ChatGPT (*e.g.*, GPT-4 *vs.* GPT-4o). Our overarching protocol comparing ChatGPT-4 to human determinations could serve as a benchmark for the ethics and execution of combining human oversight with LLMs as these algorithms become more sophisticated. Similar procedures can also be used to: 1) examine the sensitivity of generative AI platforms when screening social media content compared to other commonly used methods, such as keyword filtering, to identify when simpler or more complex analytic approaches are needed; and 2) compare generative AI's accuracy when screening social media content depending on what information is entered into the model (*e.g.*, audio transcripts, post descriptions, and objects detected) to better understand the benefits and cost to each data element in our multimodal system. Future research can also include TikTok user comments, instead of solely relying on post content and descriptions.

Moreover, future studies should compare results and content across other social media platforms to understand the generalizability of our findings, as other social media services have different demographics (*e.g.*, Facebook with older users, YouTube with a wider spread) that can offer different perspectives on topics such as e-cigarette use during pregnancy (*Zote, 2024*). Additional research using ChatGPT to examine social media data should continue to examine ChatGPT's potential limitations, ways to optimize prompts to increase ChatGPT's accuracy, and when human verification and coding is most essential.

## CONCLUSIONS

By using machine learning techniques to convert visual and audio content on TikTok into text, we were able to screen social media content for specific topics (*i.e.*, pregnancy and

vaping) using ChatGPT. We leveraged ChatGPT-4's sophisticated natural language processing abilities to manage and analyze complex multimodal data—including textual descriptions, hashtags, video content, and spoken elements from TikTok posts. The effectiveness of this GPT-based approach contrasts sharply with traditional methods, such as manual coding or machine learning models, which typically require human verification of relevant content prior to analysis. Manual coding, while detailed, is prohibitively time-consuming, particularly when dealing with large datasets that include a mix of modalities. Our findings support the capacity of ChatGPT to screen out irrelevant content within large social media datasets, enhanced by its training on diverse linguistic data. While our findings support the potential for ChatGPT to root out irrelevant content, we also identified lower levels of sensitivity compared to human coders. ChatGPT therefore has potential for streamlining large datasets to prioritize human verification of a smaller subset of more targeted content.

In practice, social media serves as an ecological context that significantly influences individuals' decisions. This influence is evident in the well-documented impact of exposure to social media tobacco marketing on tobacco use decisions. Despite regulatory attempts, trends on social media can shift rapidly. Analyzing these changes is challenging due to the vast amount of content created daily. Generative AI, such as ChatGPT, has substantial potential to facilitate social media analysis, given its ability to process large amounts of information quickly. Our case study focused on using ChatGPT-4 to screen TikTok data, converted into text, for vaping and pregnancy content due to the limited literature on this topic. Future research can apply our machine learning methods for converting large multimodal datasets into text to examine in ChatGPT or other generative AI and large language models (LLMs). Positioned as an initial screening tool, LLMs may offer greater efficiency but lower accuracy compared to human coders. More research is needed to optimize what prompts are used in generative AI platforms to increase its sensitivity when screening text-based social media content for specific topics.

## ACKNOWLEDGEMENTS

Please note that the purpose of this research was to assess the potential for generative artificial intelligence to screen social media content. As such, we used ChatGPT to screen social media content as reported in the manuscript. Additionally, we used ChatGPT to format citations and check grammar. The authors reviewed the suggested grammar edits by ChatGPT and manually edited all content as needed and take full responsibility for the content of the publication.

### Funding

This study was funded by the National Institute of Drug Abuse (NIDA) and the Food and Drug Administration's Center for Tobacco Products (FDA CTP) grant R01DA049878 (PI: Kong). Support for Drs. Kong and Ouellette was also provided by U54DA036151 (MPI:

Krishnan-Sarin, O'Malley) and support for Dr. Ouellette was provided by NIDA T32 DA019426-18 (PI: Tebes). There was no additional external funding received for this study. The funders had no role in study design, data collection and analysis, decision to publish, or preparation of the manuscript.

### Grant Disclosures

The following grant information was disclosed by the authors:
National Institute of Drug Abuse (NIDA).
Food and Drug Administration's Center for Tobacco Products (FDA CTP): R01DA049878, U54DA036151.
NIDA T32 DA019426-18.

### Competing Interests

The authors declare that they have no competing interests.

### Author Contributions

- Kellen Sharp conceived and designed the experiments, prepared figures and/or tables, authored or reviewed drafts of the article, and approved the final draft.
- Rachel R. Ouellette conceived and designed the experiments, analyzed the data, authored or reviewed drafts of the article, and approved the final draft.
- Rujula Singh Rajendra Singh conceived and designed the experiments, performed the experiments, analyzed the data, performed the computation work, prepared figures and/or tables, authored or reviewed drafts of the article, and approved the final draft.
- Elise E. DeVito conceived and designed the experiments, authored or reviewed drafts of the article, and approved the final draft.
- Neil Kamdar conceived and designed the experiments, performed the computation work, authored or reviewed drafts of the article, and approved the final draft.
- Amanda de la Noval analyzed the data, authored or reviewed drafts of the article, and approved the final draft.
- Dhiraj Murthy conceived and designed the experiments, analyzed the data, performed the computation work, authored or reviewed drafts of the article, and approved the final draft.
- Grace Kong conceived and designed the experiments, authored or reviewed drafts of the article, and approved the final draft.

### Data Availability

The dataset is available at GitHub and Zenodo:
- https://github.com/computationalmedialab/Generative-Artificial-Intelligence-and-Machine-Learning-Methods-to-Screen-Social-Media-Content
- D Murthy, knsharp, & rujulasingh. (2025). knsharp/Generative-Artificial-Intelligence-and-Machine-Learning-Methods-to-Screen-Social-Media-Content: v1.0.1 (v1.0.1). Zenodo. https://doi.org/10.5281/zenodo.14756736.

## Supplemental Information

Supplemental information for this article can be found online at http://dx.doi.org/10.7717/peerj-cs.2710#supplemental-information.

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
