# Peer review of "Generative artificial intelligence and machine learning methods to screen social media content"

_PeerJ Computer Science, doi:10.7717/peerj-cs.2710_

## Round 0.1 · original submission · Major Revisions

Dear authors,

You are advised to critically respond to all comments point by point when preparing an updated version of the manuscript and while preparing for the rebuttal letter. Please address all comments/suggestions provided by reviewers, considering that these should be added to the new version of the manuscript.

Kind regards,
PCoelho

Reviewer 1 ·

Basic reporting

This is a well-written manuscript.

This study aims to develop a multimodal system using machine learning methods (i.e., object detection, optical character recognition, and automatic speech recognition) that converts video-based content into text, which is then processed by ChatGPT to assess ChatGPT’s capacity to screen TikTok content related to e-cigarettes and pregnancy.

The authors did a good job of presenting the rationale for the study.

However, it would be better if the authors could further and clearly discuss to which stream of research this study contributed and what specific contributions and advancements it made in the discussion section. This is very important to justify the novelty and significance of this study.

Experimental design

Overall, the two research questions are well-defined, and the method is thoughtful.

I have the following question for which I want to seek clarification and justification from the authors.

The authors indicate: "Our study accounted for potential errors by sampling ten out of every 100 posts to be checked through human review by reviewing the post video and transcript for indicators or mentions of pregnancy and vaping" Why was the decision made to sample 10 posts out of every 100 rather than a different number? Is 10 enough to account for potential errors made by ChatGPT?

Validity of the findings

The findings were presented clearly.

However, I am not sure whether this conclusion is tenable: "Our findings support the capacity of ChatGPT to process large social media datasets, enhanced by its training on diverse linguistic data." This is because, among the 1,040 posts that ChatGPT marked as relevant to both pregnancy and vaping, the human coder only found 472 discussing e-cigarette use during pregnancy. This number is less than half of the entire sample.

Additional comments

I wonder whether the authors conducted any verification of the content ChatGPT determined to be irrelevant to both pregnancy and vaping. Is it possible that ChatGPT classified a TikTok post actually concerning e-cigarette use during pregnancy as not relevant to these topics? If so, were there any procedures to address this issue?

·

Basic reporting

The writing is generally clear.

There is insufficient detail in lines 391-395: how many posts in total were reviewed? how many errors? how were the errors used to prompt the models?

"Few shot learning" (line 168) generally refers to supplying a small number of labelled examples (e.g. in the prompt). Was that applied here? Is that what is happening in lines 394-395?

A little more detail in describing the data compilation (334-343) would be useful. How exactly were the components fed into ChatGPT?

The references introduce the context, but there is a lack of detail in describing the methods in previous research. In particular, what alternatives (e.g. keyword filtering) to the approach used here have been used in the past? How have such methods been evaluated? Weaknesses and failure modes of ChatGPT are discussed in the methodology but not in the introduction.

Experimental design

The paper describes an application of machine learning models to filtering social media for data collection.

Currently, this particular approach is not clearly compared to an alternative (e.g. filtering on keywords). To motivate the use of these computationally intensive methods requires demonstrating the benefits over simpler alternatives and to have a clearer picture of how much work was put into prompt engineering.

Ideally, the experiments would also investigate the benefits of each component (ASR, OCR, object detection) of the process.

The main empirical result is a precision for the classification of posts as relevant. It would be useful to have some estimate of the recall of this method, ie of all relevant posts how many are found?

Currently, I don't see any information about ethics. Was consent gained from the posters?

Validity of the findings

The main conclusion of the study is that ML+ChatGPT can screen out irrelevant content. Currently, the results do not demonstrate that, as there would need to be a comparison between irrelevant content without ChatGPT vs irrelevant content with ChatGPT. I'm fairly sure this would show what the authors claim, but at the moment the results only measure the latter.

More specific research questions may help clarify the experiments needed to answer them.

---

## Round 0.2 · Minor Revisions

Dear authors,
Thanks a lot for your efforts to improve the manuscript.
Nevertheless, some concerns are still remaining that need to be addressed.
Like before, you are advised to critically respond to the remaining comments point by point when preparing a new version of the manuscript and while preparing for the rebuttal letter.

Kind regards,
PCoelho

·

Basic reporting

The report is generally clear and the revisions have helped clarify the approach further. Thanks.

Experimental design

The revisions help clarify the experiments. Thanks.

I have a few minor remaining concerns.

Could you be clearer about the ethical review applied to the use of social media data? Was this study approved by an ethics board? If not what ethical framework or guidelines have been followed (e.g. https://aoir.org/ethics/ or https://www.hhs.gov/ohrp/sachrp-committee/recommendations/attachment-e-july-25-2022-letter/index.html)?

I am still unclear about exactly how the data was fed in to ChatGPT. My reading of the article is that a single prompt followed by a series of comma separated fields representing the metadata and text for all 11,673 videos was pasted into the input field. Is that correct?

If so, can you specify exactly what fields were fed into ChatGPT and how they were structured? It might help to give an example.

I am also unclear about the use of object detection and text recognition from the videos.

Lines 397-413 suggest that 200,436 frames were extracted from the full 11,673 videos for detection of objects and text and the output was fed into ChatGPT as text. (Was this in a separate field from the ASR output?)

Lines 528-545 suggest that 200,436 frames were extracted from the 1,040 videos identified as relevant by ChatGPT. Are these a different set of frames? Why not just use the object and text detection results already processed?

Your analysis of the objects and text commonly found in relevant posts might be more interesting if they were contrasted with those found in irrelevant posts.

Validity of the findings

The newly included results on the posts that ChatGPT identified as irrelevant make clear the benefits of this approach. Thanks.

---

## Round 0.3 · accepted · Accept

Dear authors, we are pleased to verify that you meet the reviewer's valuable feedback to improve your research.

Thank you for considering PeerJ Computer Science and submitting your work.

Kind regards
PCoelho

·

Basic reporting

no comment

Experimental design

no comment

Validity of the findings

no comment

Additional comments

My questions have been clarified. Thanks to the authors for their responses.